# Oxygen desaturation and lung ultrasonography as markers of diffuse parenchymal lung diseases severity

**Ahmed Sadaka[1]ᵒ, Asmaa Gomaa[2]ᵒ, Hoda Abdelgawad[3], Nashwa H. Abdelwahab[1], Eman Ahmed Hatata[1], Hanaa Shafiek◉[1]\***

**1** Chest diseases department, Faculty of medicine, Alexandria University, Alexandria, Egypt, **2** Chest diseases department, Students Hospital, Alexandria University, Alexandria, Egypt, **3** Cardiology department, Faculty of medicine, Alexandria University, Alexandria, Egypt

ᵒ These authors contributed equally to this work.
\* whitecoat.med@gmail.com

## Abstract

### Purpose

we aimed to evaluate lung ultrasound (LUS) and oxygen desaturation as markers for the severity of diffuse parenchymal lung disease (DPLD), specifically the fibrotic sub-types, and correlate the findings with high-resolution computed tomography (HRCT) and other physiologic parameters.

### Methods

A case-control study was conducted recruiting 31 DPLD patients and 20 age-matched healthy controls from our institution. All participants had a spirometry, HRCT, 6-minute walk test (6MWT), echocardiography and full-night cardio-respiratory polygraph. LUS for B-line quantification and pleural examination was performed on 6 zones bilaterally.

### Results

Compared to controls, patients had a statistically significant higher total number of B-lines, lower 6MWT nadir $O_2$ and lower nadir nocturnal oxygen saturation ($SpO_2$). Among patients; fibrotic DPLD (58.1%) had more B-lines, pleural irregularities with or without fragmentation, higher Warrick scores and lower 6MWT nadir $SpO_2$ (p = 0.01, 0.008, < 0.005, 0.03 respectively). There was a statistically significant positive correlation between LUS findings and Warrick score that inversely correlated with the forced vital capacity (FVC)% predicted (p < 0.001). A score of LUS findings, 6MWT nadir $SpO_2$ and time spent with $SpO_2$ < 90% (T90) ≥2 points had a sensitivity of 91.7% and specificity of 66.7% in predicting severe fibrotic DPLD (area under curve (AUC)= 0.832, CI95% = 0.723–0.941, p = 0.001).

**Data availability statement:** All relevant data are within the paper and its Supporting Information files. Further, the raw data is available in Figshare https://doi.org/10.6084/m9.figshare.28054862.v1 DataCite: SADAKA, AHMED; Gomaa, Asmaa; Abdelgawad, Hoda; Wahab, Nashwa Hassan Abdel; Shafiek, Hanaa (2025). Oxygen desaturation and lung ultrasonography as markers of diffuse parenchymal lung diseases severity. figshare. Dataset. https://doi.org/10.6084/m9.figshare.28054862.v1

**Funding:** The author(s) received no specific funding for this work.

**Competing interests:** The authors have declared that no competing interests exist.

**Abbreviation List:** DPLDs: diffuse parenchymal lung diseases; PAH: pulmonary artery hypertension; NOD: nocturnal oxygen desaturation; OSA: obstructive sleep apnea; HRCT: high-resolution computed tomography; LUS: lung ultrasound; PAP: pulmonary artery pressure; 6MWT: 6-minute walk test; $SpO_2$: nocturnal oxygen saturation; BMI: body mass index; sPAP: systolic pulmonary artery pressure; mPAP: mean pulmonary artery pressure; GGO: ground glass opacities; T90: total sleep time spent with $SpO_2 < 90\%$; SDB: sleep disordered breathing; AHI: apnea/hypopnea index; h: hour; IQR: interquartile range; SD: standard deviation; n: number; OR: odd ratio; CI95%: 95% confidence interval; ROC: receiver operating characteristic curve; AUC: area under the curve; 6MWD: 6-minute walking distance; EID: exercise induced desaturation; ILDs: interstitial lung diseases.

## Conclusions

The number of B-lines and pleural irregularities in LUS, nocturnal desaturation and exercise desaturation can play a role as markers of DPLD severity.

## Introduction

Diffuse parenchymal lung diseases (DPLDs) represent a broad spectrum of heterogeneous groups of disorders characterized by diffuse pulmonary fibrosis with variable presentations and prognoses [1]. The primary physiological derangement of DPLD is impaired gas exchange leading to hypoxemia–a crucial sequel in the natural history of DPLP– [2] that is linked to breathlessness, reduced physical activity and survival [3]. Moreover, nocturnal oxygen desaturation (NOD) occurs frequently in DPLD, whether obstructive sleep apnea (OSA) is present or not [4], and it is an independent predictor of poor prognosis [5].

Several factors are also associated with shortened survival, including older age, smoking history, more severe physiologic impairment such as annual decline in forced vital capacity (FVC) or diffusing capacity for carbon monoxide (DLCO) and in the distance walked during the 6-minute walk test (6MWT). Similarly, the development of other complications, especially pulmonary artery hypertension (PAH) and a greater radiologic extent of disease as assessed by high-resolution computed tomography (HRCT) are associated with higher mortality [6–8].

In recent years, lung ultrasound (LUS) has been studied as a useful bedside non-invasive feasible method for the evaluation of DPLD compared with HRCT [9], the gold standard radiological evaluation of DPLD [10]. Several findings associated with pulmonary fibrosis were reported during LUS assessment as thickening of the hyperechoic pleural line, an irregular and/or fragmented hyperechoic pleural line, an increased in the number (>3) of comet tail vertical artifacts (i.e., B-lines) between two ribs in a single sonographic window, and subpleural nodules [11–14].

We hypothesized that LUS and NOD can play a role as markers for the severity of DPLD. Accordingly, we aimed primarily to assess LUS utility as a diagnostic tool for DPLDs and to evaluate LUS and reduced oxygen saturation ($SpO_2$) as markers for the severity of DPLD, with emphasis on fibrotic subtypes. Further, we aimed to correlate the LUS findings with the HRCT and the physiological parameters as lung function parameters, pulmonary artery pressure (PAP), 6MWT and nocturnal $SpO_2$. Secondly, we planned to generate a score for disease severity evaluation based on the previously studied parameters.

## Methods

### Study design and ethics

This was a case–control study that enrolled 51 participants (31 DPLD patients and 20 age-matched controls) admitted to our institution from 1st September 2021–30th November 2022. All patients were adults (> 18 years old) and diagnosed with DPLD based on chest HRCT according to ATS/ERS guidelines [10,15]. Those with

comorbidities (such as heart failure (either systolic or diastolic), chronic or acute renal failure on dialysis, advanced liver cirrhosis, hepatocellular carcinoma, history of breast cancer and obesity), those associated with other pulmonary diseases (including tuberculosis, bronchiectasis, lung cancer, and a known case of obstructive sleep apnea with non-invasive therapy), those proven to have post-COVID 19 fibrosis and those who refused to sign the consent were excluded from the study. An informed written consent was obtained from all participants before their enrollment in the study which was approved by our local ethical medical committee (registration number: 0201494/ 2021).

## Participants' characteristics

All the participants underwent thorough history taking (including symptomatology, duration of illness, smoking history, occupational and environmental exposures, comorbidities, and family history of similar diseases) and full clinical examination including anthropometric measurement and body mass index (BMI).

All the patients had spirometry and 6MWT according to ATS guidelines [16,17]. Patients with FVC ≤ 50% were considered to have a severe disease [18].

A standard 2D transthoracic echocardiography was performed for all participants for the assessment of the resting systolic pulmonary artery pressure (sPAP), which was estimated by the modified Bernoulli equation [$RVSP = 4V^2 + RAP$] and mean pulmonary artery pressure ($mPAP = sPAP \times 0.61 + 2$). Those with mPAP ≥ 20 mmHg at rest [19] or sPAP ≥ 35 mmHg at rest [20,21] were considered as having PAH. A mPAP < 35 mmHg was considered as mild PAH, 35–44 mmHg as moderate PAH, and ≥ 45 mmHg as severe PAH [22]. Similarly, sPAP of 35–50 mmHg was considered as mild PAH, 50–70 mmHg as moderate PAH, and > 70 mmHg as severe PAH.

## High resolution Computed Tomography (HRCT)

All the participants underwent HRCT of the chest. HRCT characteristics were carefully identified, including: (1) upper versus lower lobar predominance, (2) mosaic attenuation, (3) peripheral subpleural or peribronchovascular involvement, (4) reticular changes, traction bronchiectasis, honeycombing or interlobular septal thickening, (5) nodular changes, and (6) other findings as hilar and/or mediastinal lymphadenopathy, consolidation changes and pleural effusion. According to the HRCT findings, the patients with DPLD were assessed qualitatively and were divided into 2 subgroups: a fibrotic group (predominant reticulations, honeycombing and traction bronchiectasis) and a non-fibrotic group (predominant ground glass opacities (GGO) and nodular pattern).

Moreover, pulmonary involvement was scored according to the semiquantitative Warrick score [13], which depends on 5 parenchymal abnormalities: ground glass opacities, irregular pleural margin, septal or subpleural lines, honeycombing and subpleural cysts that were graded into points. Warrick scoring was obtained by summing the points of the parenchymal abnormality's different patterns ranging from 0 to 15 (severity score) and the extension score was calculated by localizing the number of pulmonary segments involved for each abnormality with a maximum score of 15. The total Warrick score ranged from 0 (no involvement) to 30 (the worst involvement).

## Lung ultrasound (LUS)

All participants underwent LUS that was performed by well-trained pulmonologists (AS and AG) blinded to HRCT findings using a convex probe (3.5 MHz). The depth of penetration was standardized to 4–8 cm starting from the pleural line. The focus of the image was set at the level of the pleural line. The examination was performed while the patient was in the supine position. Each hemithorax was divided into 3 zones, which were subdivided into upper and lower regions by the parasternal, anterior and posterior axillary lines yielding 6 sonographic zones on each hemithorax.

Assessment of the pleura was done regarding abnormalities in pleural thickness (i.e., ≥ 3 mm in thickness), irregular and/or interrupted pleural surface as well as subpleural nodules [11]. Further, lung sliding (the 'to-and-fro' dynamic

movement of the lung during respiration) was estimated. In addition, B-lines (i.e., comet-tail artifacts) [14] were recognized and quantified in each zone. The B-line score was calculated as the total number of B-lines in 12 examined zones and the total number of positive chest zones score with ≥3 B-lines. The duration of LUS examination was 20–25 minutes for every participant.

### Polygraph

A cardio-respiratory screening using overnight level III device (sleep fairy device, model SF-A9, Hunan VentMed Medical Technology Co., China) was used to evaluate mainly nocturnal $SpO_2$. The device constitutes of 3 channels; nasal cannula for airflow assessment, single channel respiratory belt for respiratory movement and finger pulse oximeter to assess basal $SpO_2$, lowest $SpO_2$ (nadir $SpO_2$), average $SpO_2$, and total sleep time spent with $SpO_2 < 90\%$ (T90). Desaturation was defined as ≥3% decrease in $SpO_2$ from baseline [23]. Standard definitions for sleep disordered breathing (SDB) according to the updated American Academy of Sleep Medicine (AASM) criteria in 2012 were considered in the review of polygraphic study. Apnea/ hypopnea index (AHI) ≥5 events/hour (h) was required for the diagnosis of OSA.

### Statistical analysis

All statistical analyses were performed using IBM SPSS software package version 28.0 (Chicago, IL, USA). All the quantitative data were expressed as median and interquartile range (IQR) or mean±standard deviation (SD) according to the normal distribution of the data, while categorical variables were expressed as number (n) and percentage (%). The Kolmogorov-Smirnov test was used to verify the normality of data distribution. Mann Whitney U, independent student's t-test, Chi-square and fisher exact tests were used as appropriate. Spearman's correlation test was used to assess the correlation between the variables of interest utilizing both the data of DPLD and the control group. Logistic regression was used to estimate the odd ratio (OR) and 95% confidence interval (CI 95%) of different clinical, functional and LUS variables in relation to the severity of the disease (as defined by fibrotic DPLD disease and FVC ≤ 50%) [19] and desaturation during 6MWT. Univariate regression analysis was used initially to assess the most relevant variables and then a multivariate model was generated based on the most relevant variables using forward enter method.

A receiver operating characteristic (ROC) curve and area under the curve (AUC) analysis were used to get the best cut-off point for score generation considering severe fibrotic DPLD (positive data) versus both non-fibrotic DPLD and control groups (negative data). An LUS score was generated including 2 main items (pleural thickening and/or fragmentation, and B-lines ≥3) that had the best correlation with clinical and functional parameters as well as OR. Then, a severity score was generated that included 3 items namely the previously generated LUS score, T90 and nadir SpO2 during 6MWT. The score was calculated by the sum of all the points (present=1 and absent=0) and ranged between 0 (lowest) and 5 (highest). Further, ROC and AUC were used to assess both LUS and severity scores in predicting DPLD as a whole versus the control group. A two- tailed p-value ≤ 0.05 was considered statistically significant.

Sample size of 31 cases and 20 controls achieved 80% statistical power to detect AUC of 0.8 of LUS score in predicting severe fibrotic DPLD, and correlation coefficient of 0.7 between LUS and clinical parameters, at level of significance of 0.05 [24,25]. PASS (power analysis and sample size) software (Version 2000, NCSS, Kaysville, UT, http://www.ncss.com/software/pass) was used for power analysis calculation.

## Results

### Participants' characteristics

There was no statistically significant difference between the DPLD patients and the control group regarding age, gender, smoking history and BMI (p>0.05, S1 Table). None of the participants had cardiac comorbidities as left ventricular dysfunction, cardiomyopathies or dysrhythmias as being screened by echocardiography. However, the patients' group had

a statistically significant lower FVC, $FEV_1$, $FEV_1$/FVC, baseline $SpO_2$, 6-minute walking distance (6MWD), 6MWT nadir $SpO_2$, nadir nocturnal $SpO_2$ as well as higher T90, total number of B-lines, pleural abnormalities and Warrick score when compared to controls (p<0.01, S1 Table).

Eighteen (58.1%) had fibrotic DPLD and 13 (41.9%) had non-fibrotic DPLD. There was no statistically significant difference regarding age, gender, BMI, smoking history and PAP (p>0.05, Table 1). The most common disease was hypersensitivity pneumonitis (6 patients, 33.3%) among the fibrotic group and sarcoidosis (10 patients, 76.9%) among the non-fibrotic DPLD (p=0.014, Table 1). Spirometric parameters were significantly lower among the fibrotic group (p<0.05, Table 1) and 12 (66.7%) patients of them had severe disease based on FVC ≤ 50% [19], among whom 5 patients (41.7%) showed exercise induced desaturation (EID) during 6MWT (p=0.155, Table 1 and Fig 1). There was no statistically significant difference between the fibrotic and non-fibrotic groups regarding 6MWD and baseline $SpO_2$ (p>0.05, Table 1) but the 6MWT nadir $SpO_2$ was significantly lower among the fibrotic group (p=0.024, Table 1). The fibrotic subgroup, compared to non-fibrotic DPLD, had more total B-lines, more zones with B-lines ≥3, pleural irregularities, pleural fragmentation and higher Warrick scores (p<0.05, Table 2).

According to polygraph study, there were no detected central apneas in both patients and control groups and only infrequent obstructive events were detected with AHI 0.81 ± 1.85 events/h versus 0.02 ± 0.06 respectively (p<0.001, S1 Table). Further, there was no statistically significant difference between the fibrotic and non-fibrotic groups regarding AHI and T90 (p>0.05, Table 1) and only one patient had 9.9 events/h in the non-fibrotic subgroup (classified as mild OSA according to AASM) [23]. However, the ODI was significantly higher among the fibrotic group versus the non-fibrotic one (3.9 (2.4–6.0) vs. 1.7 (1.1–3.2), p=0.024, Table 1).

## Correlations

There was a statistically significant inverse correlation between LUS parameters (mainly total B-lines, number of zones with B-lines ≥3, pleural irregularities and fragmentation) and the FVC% predicted (r=-0.693, p<0.001; r=-0.753, p<0.001; r=-0.700, p<0.001; r=0.722, p<0.001 respectively; Fig 2 and S2 Table). Further, there was a statistically significant inverse correlation between LUS parameters and various physiological parameters (including $FEV_1$, baseline SpO2, 6MWT nadir $SpO_2$, nocturnal nadir $SpO_2$; p<0.001) and positive correlation between LUS parameters and both T90 and Warrick score (p<0.001, S2 Table). However, mPAP correlated with nocturnal nadir $SpO_2$ and T90 (p<0.01, S3 Table) but not LUS parameters of interest (p>0.05, S2 Table).

## Regression analysis

Univariate logistic regression was conducted including various demographic, physiological, and LUS parameters in relation to disease severity (S4 Table). Multivariate logistic regression model showed that LUS score was a significant predictor of DPLD severity after being corrected to age, gender, T90, nadir nocturnal SpO2, severity of PAP and desaturation during 6MWT (OR=6.63, CI95%=1.29–34.36, Table 3).

## ROC analysis

ROC and AUC analysis revealed that having ≥3 zones with B-lines ≥3 and pleural irregularity in ≥ 8 zones had a sensitivity of 91.7% and specificity of 66.7% in predicting severe fibrotic DPLD, while pleural fragmentation noted in ≥ 2 zones had a sensitivity of 66.7% and specificity of 84.6% (S1 Fig). Accordingly, LUS score was generated from 2 items (i.e., B-lines ≥3 and pleural abnormalities either irregularity or fragmentation) that had the best correlation with clinical and functional parameters, OR and AUC. The LUS score ≥ 2 had a sensitivity of 83.3% and a specificity of 66.7% in predicting severe fibrotic DPLD (AUC=0.806, CI95%=0.690–0.922, p=0.001; Fig 3A). Subsequently, 6MWT nadir $SpO_2$ ≤ 92% had a sensitivity of 94.7% and specificity of 55.6%, and T90 ≥ 6.5% had a sensitivity of 45.5% and specificity of 84.6% in predicting

**Table 1. Comparison between the fibrotic and non-fibrotic DPLD in the patients' group regarding the baseline and functional characteristics.**

| Variable | Non-Fibrotic DPLD N = 13 (41.9%) | Fibrotic DPLD N = 18 (58.1%) | p value |
|---|---|---|---|
| Age (years) | 46.46 ± 13.22 | 43.5 ± 10.06 | 0.484 |
| Gender Female/ Male | 11 (84.6)/ 2 (15.4) | 14 (77.8)/ 4 (22.2) | 0.634 |
| BMI (kg/m²) | 28.3 (24.61–29.3) | 26.07 (24.83–28.3) | 0.222 |
| Non-smoker/ smoker | 13 (100)/ 0 (0) | 15 (83.3)/ 3 (16.7) | 0.121 |
| Diagnosis (n (%)) | | | |
| HP | 0 (0) | 6 (33.3) | 0.014* |
| Sarcoidosis | 10 (76.9) | 3 (16.7) | |
| RA-ILD | 1 (7.7) | 3 (16.7) | |
| NSIP | 0 (0) | 2 (11.1) | |
| SSc-ILD | 2 (15.4) | 2 (11.1) | |
| Pneumoconiosis | 0 (0) | 2 (11.1) | |
| mPAP (mmHg) | 34 (19–46) | 23 (16–37) | 0.340 |
| sPAP (mmHg) | 27.5 (20–35) | 29 (22–33) | 0.960 |
| Severity of PAH (n (%)) | | | |
| No | 3 (23.1) | 7 (38.9) | 0.559 |
| Mild | 4 (30.8) | 7 (38.9) | |
| Moderate | 3 (23.1) | 2 (11.1) | |
| Severe | 3 (23.1) | 2 (11.1) | |
| FVC (L) | 2.6 (1.94–3.26) | 1.63 (0.65–2.07) | 0.002* |
| FVC (% predicted) | 69.0 (73.0–86.0) | 46.5 (21.0–59.0) | < 0.001* |
| FVC ≤ 50% (n (%)) | 0 (0) | 12 (66.7%) | < 0.001* |
| $FEV_1$ (L) | 1.92 (1.71–2.64) | 1.15 (1.64–0.46) | 0.001* |
| $FEV_1$ (% predicted) | 75.0 (64.0–83.51) | 44.5 (24.0–63.0) | < 0.001* |
| $FEV_1$/FVC | 84.28 (80.81–85.4) | 77.97 (71–83.2) | < 0.001* |
| $FEF_{25-75}$ (% predicted) | 78.0 (63.0–85.0) | 47.0 (33.0–75.0) | 0.001* |
| EID No/ Yes | 11 (84.6)/ 2 (15.4) | 11 (61.1)/ 7 (38.9) | 0.155 |
| 6MWD (meters) | 270.0 ± 79.01 | 270.46 ± 66.91 | 0.987 |
| Baseline $SpO_2$ (%) | 96.92 ± 2.22 | 96.17 ± 2.33 | 0.371 |
| 6MWT nadir $SpO_2$ (%) | 96 (95–98) | 93.0 (90.0–96.0) | 0.024* |
| ODI (events/ h) | 3.9 (2.4–6.0) | 1.7 (1.1–3.2) | 0.024* |
| AHI (events/ h) | 0.3 (0.1–1.5) | 0.1 (0.0–0.3) | 0.117 |
| T90 (% of sleep) | 1.0 (1.0–8.0) | 2.0 (1.0–35.0) | 0.392 |
| Nocturnal nadir $SpO_2$ (%) | 82.0 (80.0–87.0) | 79.0 (76.0–88.0) | 1.00 |

Abbreviations; n: number, BMI: body mass index, HP: hypersensitivity pneumonitis, RA-ILD: rheumatoid arthritis related interstitial lung disease, SSC-ILD: scleroderma related interstitial lung disease, NSIP: non-specific interstitial pneumonia, mPAP: mean pulmonary artery pressure, sPAP: systolic pulmonary artery pressure, PAH: pulmonary artery hypertension, FVC: forced vital capacity, L: liter, $FEV_1$: forced expired volume in 1 second, $FEF_{25-75}$: forced expiratory flow between 25–75 percentile, EID: exercise induced desaturation, $SpO_2$: oxygen saturation, 6MWD: 6-minute walk distance, 6MWT: 6-minute walk test, ODI: oxygen desaturation index, AHI: apnea/ hypopnea index, h: hour, T90: timed oxygen saturation < 90%. Qualitative data are presented as number (%) and quantitative data are presented as median (IQR) or mean ± SD according to data distribution.

* Significant p value < 0.05.

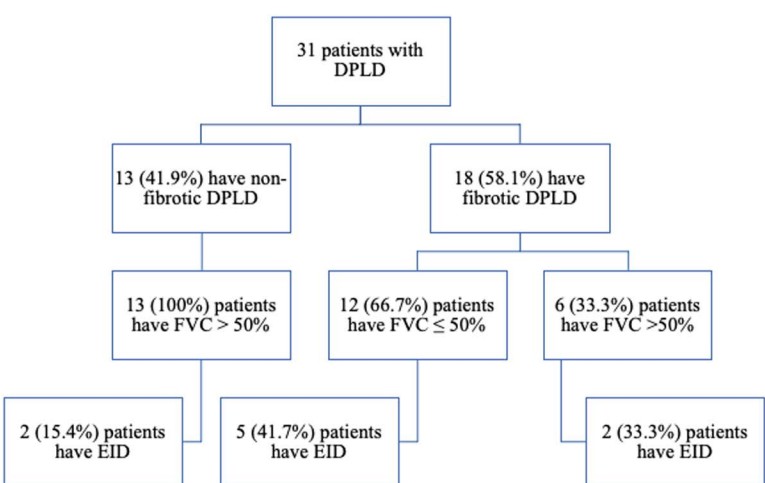

**Fig 1. Distribution of the DPLD group of patients according to HRCT evidence of fibrotic disease, FVC ≤50% and exercise induced desaturation (EID) evaluated during 6MWT.**

severe fibrotic DPLD (S2 Fig). Accordingly, a severity predicting score was generated which consisted of LUS score (2 items), 6MWT nadir $SpO_2$ and T90; and a score of ≥2 had a sensitivity of 91.7% and specificity of 66.7% in predicting severe fibrotic DPLD (AUC = 0.832, CI95% = 0.723–0.941, p = 0.001; Fig 3). Moreover, both LUS and severity predicting scores can significantly well differentiate between the DPLD and healthy control groups (AUC = 0.903, CI95% = 0.816–0.990, p < 0.001 for both LUS and severity scores, Fig 4A and 4B respectively) as a score ≥2 had a sensitivity of 74.2% and 77.4% respectively with a specificity of 100% in predicting DPLD.

## Discussion

The current study showed that LUS findings correlate well with HRCT in terms of Warrick score and pulmonary function parameters, most importantly FVC, the main severity marker used for DPLD in many studies. Further, the generated LUS score shows a high sensitivity in predicting severe fibrotic DPLD that improved on adding 6MWT nadir $SpO_2$ and T90 (sensitivity of 92%) denoting a good screening test for disease severity.

### Previous studies

Gargani et al [14] Gasperini et al [26] and Tardella et al [27] have shown that B-lines, an ultrasonographic finding in interstitial lung diseases (ILDs), strongly correlate with HRCT in their different studied populations of ILDs. Moreover, Asano et al [24] reported good correlations between B-lines in LUS and the extent of the reticular pattern on HRCT as well as FVC% predicted. Kumar et al [28] demonstrated that FVC and $FEV_1$% predicted showed a good correlation with LUS parameters and that LUS findings can provide a semi-quantitative assessment of disease severity in patients with ILD. In addition, Srivastava et al [29] found a statistically significant inverse correlation between increasing pleural line thickness in LUS and FVC% predicted, $SpO_2$ at rest and 6MWD. These results are in line with our current study.

Further, we found that DPLD group had significantly lower nocturnal nadir SpO2 compared to control group (82% vs. 92%) as well as higher T90 (13.2% vs. 0.05%) that was not associated with neither central nor obstructive respiratory events (AHI < 5 events/h in all except one patient) but higher ODI especially in the fibrotic subgroup suggesting that hypoxemia is related to the underlying lung disease rather than OSA. In accordance to our study, Yasuda et al [30] found that

**Table 2. Comparison between the fibrotic and non- fibrotic DPLD in the patients' group regarding LUS findings, CT findings and Warrick score.**

| Variable | Non-Fibrotic DPLD N = 13 (41.9%) | Fibrotic DPLD N = 18 (58.1%) | p value |
|---|---|---|---|
| **LUS findings:** | | | |
| N. of LUS zones affected | 11 (4–12) | 12 (10–12) | 0.210 |
| **Total B-lines** | 33 (11–41) | 65 (38–80) | 0.002* |
| **B lines ≥ 3 (zones)** | 5 (1–8) | 9.5 (7–11) | 0.001* |
| **Pleural thickening (zones)** | 0.0 (0.0–0.0) | 0.0 (0.0–1.0) | 0.794 |
| **Pleural irregularity (zones)** | 8.0 ± 3.81 | 10.72 ± 1.6 | 0.011* |
| **Pleural fragmentation (zones)** | 0.0 (0.0–1.0) | 2.5 (1.0–5.0) | 0.002* |
| **Limited lung sliding (zones)** | 0.0 (0.0–0.0) | 0.0 (0.0–0.0) | 0.851 |
| **Subpleural nodules (zones)** | 0.77 ± 1.01 | 1.83 ± 2.33 | 0.135 |
| **HRCT findings: (n (%))** | | | |
| **Honeycombing** | 0 (0) | 7 (38.9) | 0.011* |
| **Traction bronchiectasis** | 0 (0) | 9 (50) | 0.002* |
| **Septal lines** | 6 (61.5) | 15 (83.3) | 0.171 |
| **Subpleural cysts** | 1 (7.7) | 7 (38.9) | 0.05* |
| **Irregular pleural margin** | 3 (23.1) | 9 (50) | 0.129 |
| **GGO** | 13 (100) | 12 (66.7) | 0.02* |
| **Others** | | | |
| **Lymphadenopathy** | 8 (61.5) | 2 (11.1) | 0.003* |
| **Nodules** | 8 (61.5) | 5 (27.8) | 0.06 |
| **Consolidation** | 0 (0) | 3 (16.7) | 0.121 |
| **Pleural effusion** | 1 (7.7) | 0 (0) | 0.232 |
| **Cardiomegaly** | 2 (15.4) | 0 (0) | 0.085 |
| **Warrick score (mean ± SD)** | 8.31 ± 3.3 | 20.0 ± 4.51 | <0.001* |

Abbreviations; LUS: lung ultrasound, N: number, HRCT: high resolution CT scan, GGO: ground glass opacity. Qualitative data are presented as number (%) and quantitative data are presented as median (IQR) or mean ± SD according to data distribution.

* Significant p value < 0.05.

nocturnal nadir SpO2 was common in patients with IPF. In addition, Myall et al [31] found that nocturnal hypoxemia (T90) rather than OSA was associated with poor of quality of life and higher all-cause of mortality at 1 year in fibrotic-ILD.

## Novel findings and interpretation of the results

There have been attempts in previous studies to generate a severity score of pulmonary fibrosis based on various clinical, physiological, and radiological parameters; however, none of the scores has been unequivocally validated. Kumar et al [28] showed that a LUS score (6 items) predicts severe dyspnea in ILD group. Zhu et al [32] also generated a combined score of LUS and echocardiography (focusing on PAH) to assess ILD progression. Earlier in 2014, Ryerson et al [33] developed the ILD-GAP model (gender-age-physiology and ILD subtype) to predict disease severity and mortality but without including lung morphology. Zhu et al [34] combined the ILD-GAP score and LUS/echocardiography (right ventricular function) scores and found it is better in reflecting the severity of ILD than each score alone.

We generated a simplified score composed of LUS score (2 items), that was a significant predictor of DPLD disease severity in multivariable regression analysis (table 3), plus 6MWT nadir $SpO_2 ≤ 92\%$ plus T90 ≥ 6.5% achieving a sensitivity of 91.7% and specificity of 66.7% in predicting severe fibrotic DPLD (based on both HRCT and FVC < 50%) as well as discriminating DPLD from healthy lungs with a sensitivity of 74.2% and specificity of 100%. We suppose that the current

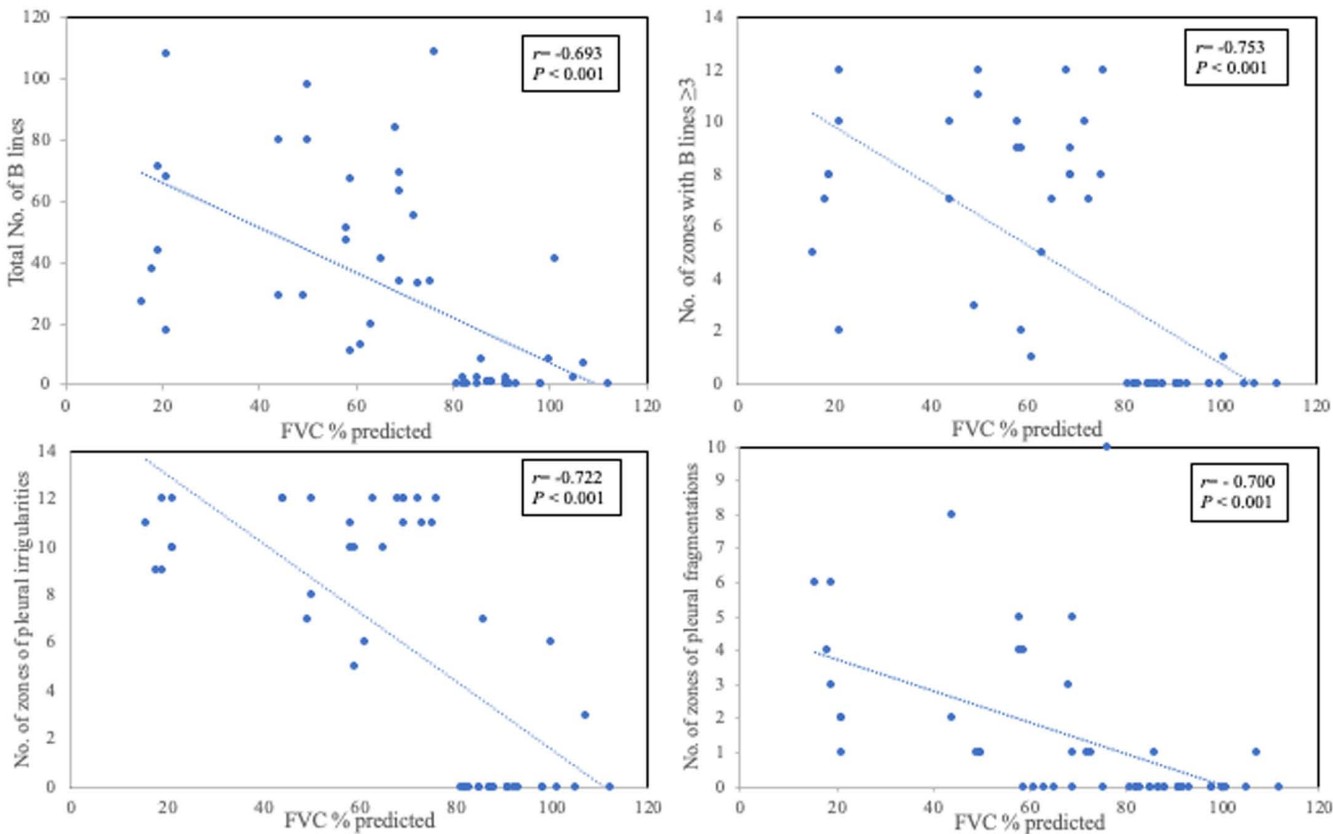

**Fig 2. Correlations between US parameters (including total B-lines (A), number of zones with B-lines ≥ 3 (B), pleural irregularities (C) and fragmentation (D)) and the FVC% predicted (r = -0.693, p < 0.001; r = -0.753, p < 0.001; r = -0.700, p < 0.001; r = 0.722, p < 0.001 respectively).**

**Table 3. Multivariate logistic regression regarding the severity of FVC (as ≤ 50% considered severe disease).**

| | B | S.E. | Sig. | OR | 95% CI for OR | |
| --- | --- | --- | --- | --- | --- | --- |
| | | | | | Lower | Upper |
| Gender | -0.07 | 1.36 | 0.96 | 0.94 | 0.07 | 13.54 |
| Age | -1.48 | 0.81 | 0.07* | 0.23 | 0.05 | 1.11 |
| T90 | -0.70 | 0.45 | 0.12 | 0.50 | 0.21 | 1.20 |
| Nadir nocturnal SpO$_2$ | 1.12 | 0.74 | 0.13 | 3.07 | 0.72 | 13.07 |
| Score LUS | 1.89 | 0.84 | 0.024* | 6.63 | 1.28 | 34.36 |
| Severity of PAH | -0.74 | 0.56 | 0.19 | 0.48 | 0.16 | 1.44 |
| EID | 2.55 | 1.67 | 0.13 | 12.74 | 0.48 | 338.42 |
| Constant | -2.76 | 2.59 | 0.29 | 0.06 | | |

Abbreviations; OR: odd ratio, 95%CI: 95% confidence interval, T90: timed oxygen saturation < 90%, SpO$_2$: oxygen saturation, LUS: lung ultrasound, PAH: pulmonary artery hypertension, EID: exercise induced desaturation.

* Significant p value < 0.05.

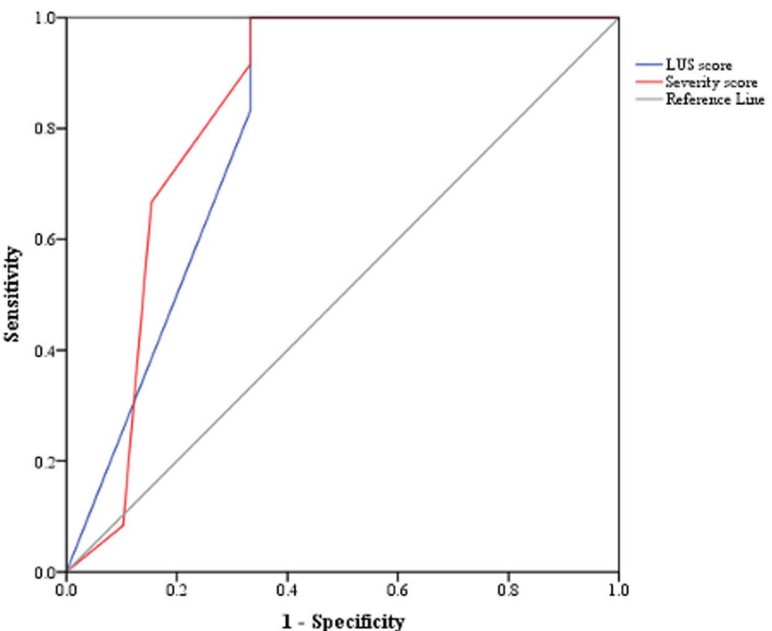

**Fig 3. ROC curve and AUC analysis for predicting severe fibrotic DPLD.** LUS score constitutes pleural abnormalities and number of zones with B-lines ≥ 3 (AUC = 0.806, CI95% = 0.690–0.922, p = 0.001) and severity score constituting of LUS score, T90 and 6MWT nadir SpO$_2$ (AUC = 0.832, CI95% = 0.723–0.941, p = 0.001) for predicting severe fibrotic DPLD (i.e., those with evidence of HRCT fibrosis and FVC ≤ 50%).

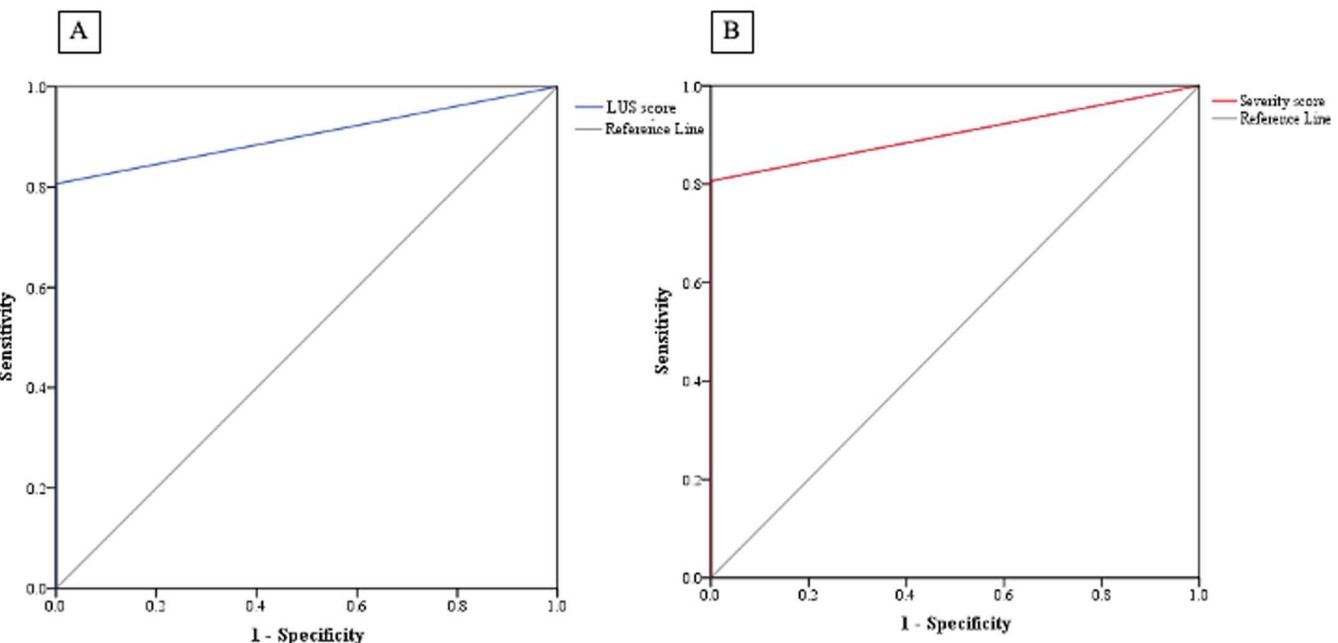

**Fig 4. A. ROC curve and AUC analysis of LUS score for predicting DPLD** (fibrotic and non-fibrotic) (AUC = 0.903, CI95% = 0.816–0.990, p < 0.001); B. ROC curve and AUC analysis of severity score for predicting DPLD (fibrotic and non-fibrotic) (AUC = 0.903, CI95% = 0.816–0.990, p < 0.001).

score is simple enough and includes different physiological/radiological parameters based on strong evidence; as both desaturation during 6MWT and NOD are important predictors of deterioration, prognosis and mortality in DPLD patients [3,31].

Moreover, we observed correlation between nocturnal $SpO_2$, T90 and mPAP. NOD is an important risk factor for PAH [8] and once it manifests, it is frequently accompanied with poorer outcomes, oxygen requirement, and functional status [34]. Our results may support the need to investigate the role of nocturnal $O_2$ therapy in DPLD even in patients who do not have daytime desaturation.

### Clinical implication

LUS is an inexpensive, bed-side and non-ionizing diagnostic tool that has been emerged recently in the evaluation of various thoracic diseases and could be applied for rapid out-patient screening in the hands of well-trained pulmonologists [35]. Based on the current results, the generated LUS score could be used as first-line screening tool for detecting DPLD by pulmonologists especially when HRCT is not available or in resource-limited areas. Further, the generated severity score (LUS and desaturation indices) is quite simple and feasible score for follow-up of DPLD progression with a sensitivity of 92% which could be more sensitive than HRCT alone for early detection of subtle worsening fibrotic-DPLD with less radiation exposure hazards [36]. In addition, the score may be more applicable in detecting progressive DPLD than spirometry which is an effort dependent test carrying significant variability between different labs. However, further studies are still recommended to evaluate the generated scores.

### Limitations

The current study has some limitations. Firstly, LUS cannot assess interstitial lesions in the deep lung tissue and cannot distinguish between the different radiological DPLD patterns (such as GGO, reticulation, honeycombing). Thus, LUS can not be considered as definite diagnostic tool of specific DPLD but it correlates well with HRCT findings as we demonstrated in our study and previous ones [14,26,27]. Secondly, we did not include DLCO in the evaluation of our participants due to its unavailability of this test in our institute. However, FVC is widely accepted as main parameter for disease monitoring in clinical trials [19]. Thirdly, our studied population is heterogenous with various causes of DPLD rather than a single pathology. Fourthly, the sample size of the current study is relatively small, however, it achieved a power analysis of 80% statistical power as aforementioned in the methodology denoting confident acceptable results [24,25]. Lastly, we did not assess the right ventricular volume and ejection fraction during echocardiographic evaluation of the studied population. However, in the current study, only 32% (10 patients) of the DPLD group had moderate-to-severe PAH. Despite the acceptability of 2D echocardiography as first tool for right ventricular evaluation, it has its limitation in patients with underlying lung diseases and the complex 3D structure of the right ventricle rendering cardiac MRI as the current gold-standard technique for functional and anatomical evaluation of right ventricle [37].

### Conclusions

Our results highlight the role of LUS in the evaluation of DPLD and its diagnostic role in identifying severe fibrotic DPLD. The number of B-lines and pleural fragmentation in LUS can be utilized as markers of disease severity. Moreover, nocturnal desaturation and exercise desaturation are simple important parameters for predicting DPLD severity and offer better reflection of disease severity when combined with LUS compared to LUS alone.

### Supporting information

**S1 File. Supplemental material. The file contains S1-S4 Tables.**
(DOCX)

**S1 Fig. ROC curve and AUC analysis of most relevant LUS parameters in predicting severe fibrotic DPLD (i.e., those with evidence of HRCT fibrosis and FVC ≤ 50%); A.** Number of zones with B-lines ≥ 3 (AUC = 0.812, CI 95% = 0.696–0.928, p = 0.001), B. Number of zones with pleural fragmentation (AUC = 0.862, CI 95% = 0.764–0.961, p < 0.001), C. Number of zones with pleural irregularity (AUC = 0.790, CI 95% = 0.668–0.911, p = 0.003). (TIF)

**S2 Fig. ROC curve and AUC analysis of 6MWT nadir SpO2** (A. AUC = 0.905, CI 95% = 0.819–0.991, p < 0.001) and T90 (B. AUC = 0.769, CI 95% = 0.622–0.917, p = 0.007) in predicting severe fibrotic DPLD (i.e., those with evidence of HRCT fibrosis and FVC ≤ 50%). (TIF)

## Author contributions

**Conceptualization:** Hoda Abdelgawad, Nashwa H. Abdelwahab, Eman Ahmed Hatata, Hanaa Shafiek.

**Data curation:** Ahmed Sadaka, Asmaa Gomaa, Nashwa H. Abdelwahab, Eman Ahmed Hatata, Hanaa Shafiek.

**Formal analysis:** Ahmed Sadaka, Asmaa Gomaa, Hanaa Shafiek.

**Investigation:** Ahmed Sadaka, Asmaa Gomaa, Hoda Abdelgawad, Hanaa Shafiek.

**Methodology:** Ahmed Sadaka, Asmaa Gomaa, Hoda Abdelgawad, Nashwa H. Abdelwahab, Eman Ahmed Hatata, Hanaa Shafiek.

**Resources:** Asmaa Gomaa.

**Supervision:** Ahmed Sadaka, Hoda Abdelgawad, Nashwa H. Abdelwahab, Eman Ahmed Hatata, Hanaa Shafiek.

**Validation:** Hoda Abdelgawad, Nashwa H. Abdelwahab, Eman Ahmed Hatata, Hanaa Shafiek.

**Visualization:** Ahmed Sadaka, Hoda Abdelgawad, Nashwa H. Abdelwahab, Eman Ahmed Hatata, Hanaa Shafiek.

**Writing – original draft:** Ahmed Sadaka, Asmaa Gomaa.

**Writing – review & editing:** Nashwa H. Abdelwahab, Eman Ahmed Hatata, Hanaa Shafiek.

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
