## [Decision Letter · Decision Letter 0]

4 Nov 2024

PONE-D-24-24333Oxygen desaturation and lung ultrasonography as markers of diffuse parenchymal lung diseases severityPLOS ONE

Dear Dr. Shafiek,

Thank you for submitting your manuscript to PLOS ONE. After careful consideration, we feel that it has merit but does not fully meet PLOS ONE’s publication criteria as it currently stands. Therefore, we invite you to submit a revised version of the manuscript that addresses the points raised during the review process.

**ACADEMIC EDITOR: ** All issues raised by expert reviewer are required.

We look forward to receiving your revised manuscript.

Kind regards,

Vincenzo Lionetti, M.D., PhD

Academic Editor

PLOS ONE

Journal Requirements: When submitting your revision, we need you to address these additional requirements. 1. Please ensure that your manuscript meets PLOS ONE's style requirements, including those for file naming. The PLOS ONE style templates can be found at https://journals.plos.org/plosone/s/file?id=wjVg/PLOSOne_formatting_sample_main_body.pdf and https://journals.plos.org/plosone/s/file?id=ba62/PLOSOne_formatting_sample_title_authors_affiliations.pdf 2. In the online submission form, you indicated that "All relevant data are within the manuscript and its Supporting Information files. Further, the raw data is available on request from the corresponding author" All PLOS journals now require all data underlying the findings described in their manuscript to be freely available to other researchers, either 1. In a public repository, 2. Within the manuscript itself, or 3. Uploaded as supplementary information.This policy applies to all data except where public deposition would breach compliance with the protocol approved by your research ethics board. If your data cannot be made publicly available for ethical or legal reasons (e.g., public availability would compromise patient privacy), please explain your reasons on resubmission and your exemption request will be escalated for approval. 3. Please include captions for your Supporting Information files at the end of your manuscript, and update any in-text citations to match accordingly. Please see our Supporting Information guidelines for more information: http://journals.plos.org/plosone/s/supporting-information.

Reviewers' comments:

Reviewer's Responses to Questions

**Comments to the Author**

1. Is the manuscript technically sound, and do the data support the conclusions?

Reviewer #1: Partly

Reviewer #2: Yes

2. Has the statistical analysis been performed appropriately and rigorously? 

Reviewer #1: Yes

Reviewer #2: Yes

3. Have the authors made all data underlying the findings in their manuscript fully available?

Reviewer #1: No

Reviewer #2: Yes

4. Is the manuscript presented in an intelligible fashion and written in standard English?

Reviewer #1: Yes

Reviewer #2: Yes

5. Review Comments to the Author

Reviewer #1: The manuscript investigates the use of lung ultrasonography (LUS) and oxygen desaturation parameters as markers for assessing the severity of diffuse parenchymal lung diseases (DPLD). The study includes 31 patients with DPLD and 20 healthy controls and proposes a scoring system based on LUS findings, 6-minute walk test (6MWT) nadir SpO2, and T90 to predict severe fibrotic DPLD.

While the study addresses an important area in pulmonology, some aspects of the manuscript need clarification and improvement

- The use of abbreviations such as DPLD, HRCT, FVC, and T90 in the abstract should be defined when first mentioned.

- The purpose of the study in the abstract and introduction is vague. It mentions LUS as a diagnostic tool and marker of severity, but it is not specified for which pathology (DPLD in this case) until later in the text. The purpose should be clearly stated as evaluating LUS and desaturation as markers for the severity of DPLD, with emphasis on fibrotic subtypes.

- The role of the control group is not well-explained. While the control group is used for baseline comparisons, its specific function is underdeveloped. Further clarification is needed on how the control group data were utilized, particularly in the context of generating diagnostic ROC curves.

- The manuscript should explain which specific groups were compared when generating the ROC curves. Currently, the ROC analysis focuses on the sensitivity and specificity for predicting DPLD severity (fibrotic DPLD) but does not assess the presence of DPLD as a whole. The distinction between the use of the control group and subgroups within the DPLD cohort should be made explicit.

- The study's sample size of 31 patients and 20 controls is relatively small. The authors should include a power analysis to justify the sample size and confirm that it is adequate to detect statistically significant differences. This would enhance the credibility and robustness of the conclusions.

- The statement “Our results support the role of LUS in the evaluation and prognostication of DPLD” is not fully supported by the data. The findings mainly highlight the diagnostic role of LUS in identifying severe fibrotic DPLD but do not demonstrate prognostic utility.

- In Figure 2, fit lines for the correlation plots should be included

- The manuscript contains sections that require refinement for clearer and more concise English. Phrasing is awkward in parts, and redundancy should be minimized, particularly in the introduction and discussion. For example, the introduction could be streamlined to focus more directly on the study's relevance and aims without excessive background information.

Reviewer #2: This study provides meaningful insights into the application of lung ultrasound (LUS) as a diagnostic tool and potential severity marker in patients with diffuse parenchymal lung disease (DPLD). The correlation of LUS findings with oxygen desaturation and HRCT scoring highlights the potential utility of LUS in clinical practice. I have some suggestions to improve the overall quality of this work:

- It would be helpful if all acronyms were spelled out when first mentioned in the abstract and main text. This enhances accessibility, especially for readers less familiar with these terms.

- The sample size may limit the generalizability of findings. Including this limitation briefly in the abstract and discussing it in more depth in the discussion section would enhance transparency. Small samples can impact the reliability of correlation and sensitivity analyses, so acknowledging this would strengthen the paper’s credibility.

- Figures could benefit from higher resolution, clearer labeling, and improved contrast. Enhancing the quality of visuals will allow readers to interpret the data more effectively.

- While echocardiography is included, there is limited detail on right ventricular function evaluation. Given its relevance to DPLD, a discussion on the limitations of assessing right ventricular function would strengthen the analysis.

- The study includes cardio-respiratory polygraphy but does not clearly distinguish between central and obstructive apneas. This is a potential limitation, as it may affect the interpretation of desaturation events, and addressing this in the discussion would improve the study’s thoroughness.

- The study does not utilize biomarkers (e.g., NT-proBNP) to differentiate cardiogenic from non-cardiogenic B-lines, which may limit the accuracy of LUS findings in identifying DPLD severity.

- Given the promising sensitivity and specificity of the combined LUS and desaturation score, additional commentary on how this could be applied in clinical settings (e.g., as a first-line screening tool in resource-limited areas) would be valuable.

6. PLOS authors have the option to publish the peer review history of their article (what does this mean? ). If published, this will include your full peer review and any attached files.

**Do you want your identity to be public for this peer review?** For information about this choice, including consent withdrawal, please see our Privacy Policy .

Reviewer #1: No

Reviewer #2: No

---

## [Author Response · Author response to Decision Letter 0]

18 Dec 2024

Reviewer #1:

Thank you for your valuable evaluation and your time.

1- The use of abbreviations such as DPLD, HRCT, FVC, and T90 in the abstract should be defined when first mentioned.

Response: We apologize for this convince that also has been raised by reviewer #2. We described the abbreviations when first mentioned in the abstract and whole manuscript.

2- The purpose of the study in the abstract and introduction is vague. It mentions LUS as a diagnostic tool and marker of severity, but it is not specified for which pathology (DPLD in this case) until later in the text. The purpose should be clearly stated as evaluating LUS and desaturation as markers for the severity of DPLD, with emphasis on fibrotic subtypes.

Response: We agree with the reviewer that the purpose of the study was not clear. Accordingly, we clarified the purpose in both the abstract and the introduction as suggested by the reviewer. [Abstract, Page 2, lines 18-20; introduction, pages 4 (lines 66-67) and page 5 (lines 120-121)]

3- The role of the control group is not well-explained. While the control group is used for baseline comparisons, its specific function is underdeveloped. Further clarification is needed on how the control group data were utilized, particularly in the context of generating diagnostic ROC curves.

Response: We agree with the reviewer that the role of the control is not well-explained in the text. Firstly, we included a control group in order to assess the diagnostic role of LUS in identification of DLPD (as part of the study purposes; page 4-last line) where there was statistically significant difference between both DPLD and control groups despite infrequent B-lines seen in the control group (p < 0.001, table S1, supplemental file). Secondly, we were looking for identifying baseline nocturnal oxygen saturation (NOD) in matched control group due to lack of similar data in our community that is still in accordance with a study in USA (Gries RE, Brooks LJ. Normal oxyhemoglobin saturation during sleep. How low does it go? Chest. 1996 Dec;110(6):1489-92). Further, the data of the control was used in the correlation between various parameters of interest (e.g., LUS findings, spirometric data and Warrick score) in conjugation with the disease group following previous studies (Cosío BG, et al. Structure-function relationship in COPD revisited: an in vivo microscopy view. Thorax. 2014 Aug;69(8):724-30; Siena L, et al. Reduced apoptosis of CD8+ T-lymphocytes in the airways of smokers with mild/moderate COPD. Respir Med. 2011 Oct;105(10):1491-500) as well as in regression model and the generation of the diagnostic ROC curves considering the data of the severe fibrotic DPLD (as positive data) versus both the non-fibrotic DPLD and control (as negative data). We clarified this point in the methods (the statistical analysis section, page 9, lines 210 and 217-218).

4- The manuscript should explain which specific groups were compared when generating the ROC curves. Currently, the ROC analysis focuses on the sensitivity and specificity for predicting DPLD severity (fibrotic DPLD) but does not assess the presence of DPLD as a whole. The distinction between the use of the control group and subgroups within the DPLD cohort should be made explicit.

Response: We agree with the reviewer that this point is missing and needs explanation. As we explained in previous point raised by the same reviewer, while generating the diagnostic ROC curves, we considered the data of severe fibrotic DPLD (as positive data) versus both the non-fibrotic DPLD and control (as negative data). However, we tested the fibrotic DPLD versus the non-fibrotic DLPD only; and we found that LUS score ≥ 2 has a sensitivity of 88.9% (AUC= 0.701, CI 95%= 0.500 – 0.902) and severity score ≥ 2 had sensitivity of 94.5% (AUC= 0.703, CI 95%= 0.497 – 0.908) in predicting the fibrotic DPLD (data not included) that still supporting the high sensitivity of both scores. We clarified this point in the methods. (Page 9, 217-218 and 223-224)

Further, we agree with the reviewer that it was missing providing the sensitivity and specificity of LUS and severity scores to assess the presence of DPLD as a whole versus healthy control group. Accordingly, and following the reviewer’s suggestion, we generated ROC analysis testing the presence of DPLD as a whole versus control group using the same generated scores described in the text (methods page 9, results pages 11-12 and online supplement); we found that both LUS and severity scores (LUS parameters and desaturation indexes) can significantly differentiate between the DPLD and healthy control groups (AUC= 0.903, CI 95%= 0.816 – 0.990, p < 0.001 for both scores) as a score ≥ 2 had a sensitivity of 74.2% and 77.4% for LUS and severity scores respectively with a specificity of 100% for both scores in predicting DPLD as a whole (either fibrotic or non-fibrotic). We included this data in the results section (Page 13, lines 304-307) with adding of the figure as figure 3B.

5- The study's sample size of 31 patients and 20 controls is relatively small. The authors should include a power analysis to justify the sample size and confirm that it is adequate to detect statistically significant differences. This would enhance the credibility and robustness of the conclusions.

Response: We agree with the reviewer that the sample size is relatively small; however, diffuse parenchymal lung diseases are relatively uncommon and highly variable worldwide (Kaul B, et al. Variability in Global Prevalence of Interstitial Lung Disease. Front Med (Lausanne). 2021;8:751181) with an average of 7% of the general population (Hunninghake GM, et al. MUC5B promoter polymorphism and interstitial lung abnormalities. N Engl J Med 2013; 368: 2192–2200; Bhattacharyya P, et al. The increasing trend and the seasonal variation in attendance of diffuse parenchymal lung disease patients presenting to a pulmonary clinic in Eastern India. Lung India. 2021 Nov-Dec;38(6):529-532).

Further, our sample size is based on the prevalence of the DPLD and interstitial lung diseases previously published in our community (Shafiek H, et al. Transbronchial cryobiopsy validity in diagnosing diffuse parenchymal lung diseases in Egyptian population. J Multidiscip Healthc. 2019 Aug 30;12:719-726; El-Hoffy MM, et al. High resolution multi-detector row computed tomography in imaging of interstitial lung diseases. Alexandria J Med. 2008;44(2):1–7; Essam H, et al. Effects of different exercise training programs on the functional performance in fibrosing interstitial lung diseases: A randomized trial. PLoS One. 2022 May 26;17(5):e0268589).

Following the reviewer’s suggestion, we calculated the power analysis (calculated using power analysis and sample size software (PASS) version 2000, www.ncss.com) in the methodology to justify the sample size and the adequality of the current statistical differences. We found that the sample size of 31 cases and 20 controls achieve 80% statistical power to detect AUC of 0.8 of LUS score for predicting severe fibrotic DPLD, and correlation coefficient of 0.7 between LUS and clinical parameters, at level of significance of 0.05. (Asano M, et al. Validity of Ultrasound Lung Comets for Assessment of the Severity of Interstitial Pneumonia. J Ultrasound Med 2018; 37(6):1523-1531; Chow SC, et al. Sample Size Calculations in Clinical Research (Second Edition); 2007.) Accordingly, the current sample size reaches a confident acceptable power rendering the good reproducibility and credibility of the current analysis. We included this part in the methods, statistical analysis section. (Page 9, lines 226-230, references 24-25)

6- The statement “Our results support the role of LUS in the evaluation and prognostication of DPLD” is not fully supported by the data. The findings mainly highlight the diagnostic role of LUS in identifying severe fibrotic DPLD but do not demonstrate prognostic utility.

Response: We agree with the reviewer that this is a mistake and the sentence is not well expressed. We corrected the sentence into “Our results highlight the role of LUS in the evaluation of DPLD and its diagnostic role in identifying severe fibrotic DPLD.” (Page 17 conclusion section, lines 433-434)

7- In Figure 2, fit lines for the correlation plots should be included.

Response: Following the reviewer’s suggestion, we included fit lines for the correlation plots. (Corrected figure 2)

8- The manuscript contains sections that require refinement for clearer and more concise English. Phrasing is awkward in parts, and redundancy should be minimized, particularly in the introduction and discussion. For example, the introduction could be streamlined to focus more directly on the study's relevance and aims without excessive background information.

Response: We agree with the reviewer that the introduction and discussion were in parts redundant and contain excessive background information. Accordingly, we re-phrased the introduction with focus on the data relevant to the study with removal of the excessive background information. (Introduction section, page 4, 1st to 3rd paragraphs) In addition, we reviewed the discussion and removed paragraphs containing excessive scientific information (page 15), re-phrased some sentences (page 15, lines 357-364), and adding others highlighting our study results and suggested by reviewer #2 (page 14, lines 338-345; and page 16, clinical implications subtitle, lines 397 - 407).

Reviewer #2:

Thank you for your valuable evaluation and your time

1- It would be helpful if all acronyms were spelled out when first mentioned in the abstract and main text. This enhances accessibility, especially for readers less familiar with these terms.

Response: We apologize for this convince that also has been raised by reviewer #1. We described the abbreviations when first mentioned in the abstract and whole manuscript.

2- The sample size may limit the generalizability of findings. Including this limitation briefly in the abstract and discussing it in more depth in the discussion section would enhance transparency. Small samples can impact the reliability of correlation and sensitivity analyses, so acknowledging this would strengthen the paper’s credibility.

Response: We agree with the reviewer that the sample size is important and small sample may limit the generalizability of findings. However, diffuse parenchymal lung diseases are relatively uncommon and highly variable worldwide (Kaul B, et al. Variability in Global Prevalence of Interstitial Lung Disease. Front Med (Lausanne). 2021;8:751181) with an average of 7% of the general population (Hunninghake GM, et al. MUC5B promoter polymorphism and interstitial lung abnormalities. N Engl J Med 2013; 368: 2192–2200; Bhattacharyya P, et al. The increasing trend and the seasonal variation in attendance of diffuse parenchymal lung disease patients presenting to a pulmonary clinic in Eastern India. Lung India. 2021 Nov-Dec;38(6):529-532).

Further, our sample size is based on the prevalence of the DPLD and interstitial lung diseases in our community (Shafiek H, et al. Transbronchial cryobiopsy validity in diagnosing diffuse parenchymal lung diseases in Egyptian population. J Multidiscip Healthc. 2019 Aug 30;12:719-726; El-Hoffy MM, et al. High resolution multi-detector row computed tomography in imaging of interstitial lung diseases. Alexandria J Med. 2008;44(2):1–7; Essam H, et al. Effects of different exercise training programs on the functional performance in fibrosing interstitial lung diseases: A randomized trial. PLoS One. 2022 May 26;17(5):e0268589).

However, we calculated the power analysis (using PASS software version 2000; www.ncss.com) and included it in the methodology as being also suggested by reviewer #1; we found that the sample size of 31 cases and 20 controls achieve 80% statistical power to detect AUC of 0.8 of LUS score for predicting severe fibrotic DPLD, and correlation coefficient of 0.7 between LUS and clinical parameters, at level of significance of 0.05. (Asano M, et al. Validity of Ultrasound Lung Comets for Assessment of the Severity of Interstitial Pneumonia. J Ultrasound Med 2018; 37(6):1523-1531; Chow S-C, et al. Sample Size Calculations in Clinical Research (Second Edition); 2008.) Accordingly, the current sample size reaches an adequate, acceptable and confident power rendering the reproducibility and credibility of the current analysis. We added this part in the methods, statistical analysis section. (Page 9, lines 226-230, references 24-25) In addition, we included a comment on the sample size in the limitations of the study in order to strengthen the paper’s credibility despite our average power analysis. (Page 17, lines 421-424)

3- Figures could benefit from higher resolution, clearer labeling, and improved contrast. Enhancing the quality of visuals will allow readers to interpret the data more effectively.

Response: Following the reviewer’s suggestion, we informed the figure resolution, labeling and contrast. Also, we added fit lines for the correlations in figure as being suggested by reviewer #1. (Corrected figures 1-3).

4- While echocardiography is included, there is limited detail on right ventricular function evaluation. Given its relevance to DPLD, a discussion on the limitations of assessing right ventricular function would strengthen the analysis.

Response: We agree with the reviewer that there is limited detail on right ventricular function evaluation. Following the reviewer’s suggestion, we included this limitation in the discussion: “we did not assess the right ventricular volume and ejection fraction during 2D echocardiography evaluation of the studied population”. However, we had only 10 patients (32.3%) had moderate to severe pulmonary hypertension without statistically significant difference between the fibrotic and the non-fibrotic subgroups (4 patients (22%) vs. 6 patients (46%), p= 0.559, table 1) even higher PAP was detected among the non-fibrotic group. Further, we did not find evidence of right heart failure in our studied population. 2D echocardiography is considered the 1st tool of right ventricle evaluation that can be limited due to the underlying lung disease and the complex 3D structure of the right ventricle, so cardiac MRI is the current gold-standard technique for functional and anatomical evaluation of right ventricle (Simon MA. Assessment and treatment of right ventricular failure. Nature Reviews Cardiology. 2013;10(4):204-18). This point is highlighted this point in the discussion, limitations section (Page 17, lines 424-430) supported by reference 37.

5- The study includes cardio-respiratory polygraphy but does not clearly distinguish between central and obstructive apneas. This is a potential limitation, as it may affect the interpretation of desaturation events, and addressing this in the discussion would improve the study’s thoroughness.

Response: We agree with the reviewer that this point is missing and is not described well in the results of the study. No central apneas were detected in the cardio-respiratory polygraph study of the studied population. All the events detected were infrequent obstructive events (< 5 events/ hour) with only 1 patient (3.2%) had 9.9 events / hour classified as mild OSA according to AASM criteria. (Berry RB, et al. Rules for scoring respiratory events in sleep: update of the 2007 AASM Manual for the Scoring of Sleep and Associated Events. Deliberations of the Sleep Apnea Definitions Task Force of the American Academy of Sleep Medicine. Journal of clinical sleep medicine : JCSM : official publication of the American Academy of Sleep Medicine. 2012;8(5):597-619).

However, ODI is significantly higher among the fibrotic group versus the non-fibrotic one (median of 3.9 (2.4 – 6.0) vs. 1.7 (1.1 – 3.2), p= 0.024, Table 1) that is associated with higher T90 and lower nadir nocturnal desaturation despite being insignificant (table 1) highlighting that all the desaturation detected were mostly due to the underlying lung diseases (i.e., DPLD) rather than associated OSA. In accordance to our data, Yasuda et al found that nocturnal nadir SpO2 was common in patients with IPF (Yasuda Y, et al. Analysis of noctu

---

## [Decision Letter · Decision Letter 1]

27 Jan 2025

PONE-D-24-24333R1Oxygen desaturation and lung ultrasonography as markers of diffuse parenchymal lung diseases severityPLOS ONE

Dear Dr. Shafiek,

Thank you for submitting your manuscript to PLOS ONE. After careful consideration, we feel that it has merit but does not fully meet PLOS ONE’s publication criteria as it currently stands. Therefore, we invite you to submit a revised version of the manuscript that addresses the points raised during the review process.

**ACADEMIC EDITOR:** the authors should carefully address the statistical issues raised by expert reviewer. Moreover, they should discuss their results in the light of previous unmentioned study (doi: 10.1097/01.CCM.0000287525.03140.3F).

We look forward to receiving your revised manuscript.

Kind regards,

Vincenzo Lionetti, M.D., PhD

Academic Editor

PLOS ONE

Journal Requirements:

Reviewers' comments:

Reviewer's Responses to Questions

**Comments to the Author**

1. If the authors have adequately addressed your comments raised in a previous round of review and you feel that this manuscript is now acceptable for publication, you may indicate that here to bypass the “Comments to the Author” section, enter your conflict of interest statement in the “Confidential to Editor” section, and submit your "Accept" recommendation.

Reviewer #1: (No Response)

Reviewer #2: All comments have been addressed

2. Is the manuscript technically sound, and do the data support the conclusions?

Reviewer #1: (No Response)

Reviewer #2: Yes

3. Has the statistical analysis been performed appropriately and rigorously? 

Reviewer #1: (No Response)

Reviewer #2: Yes

4. Have the authors made all data underlying the findings in their manuscript fully available?

Reviewer #1: Yes

Reviewer #2: Yes

5. Is the manuscript presented in an intelligible fashion and written in standard English?

Reviewer #1: Yes

Reviewer #2: Yes

6. Review Comments to the Author

Reviewer #1: I appreciate the effort in revising the paper, however thereare still some issues in the reporting of the results:

-In the new Figure 3B the ROC curve of LUS score is not visible, moreover the authors state "Moreover, both LUS and severity predicting scores can significantly well differentiate between the DPLD and healthy control groups (AUC= 0.903, CI95%= 0.816 – 0.990, p < 0.001 for both LUS and severity scores, Fig 3B) as a score ≥ 2 had a sensitivity of 74.2% and 258 77.4% respectively with a specificity of 100% in predicting DPLD."

This is not clear: do the severity score and LUS have the same AUC? then why the ROC of LUS is not reported in the figure?

-I strongly encourage the authors to revise the manuscript with the help of a native English speaker, as many sections are still difficult to read and appear verbose or convoluted.

Reviewer #2: The authors have adequately addressed the points raised and the overall quality of the manuscript is improved

7. PLOS authors have the option to publish the peer review history of their article (what does this mean? ). If published, this will include your full peer review and any attached files.

**Do you want your identity to be public for this peer review?** For information about this choice, including consent withdrawal, please see our Privacy Policy .

Reviewer #1: No

Reviewer #2: **Yes: ** Francesco Gentile

---

## [Author Response · Author response to Decision Letter 1]

8 Mar 2025

Reviewer #1: I appreciate the effort in revising the paper, however thereare still some issues in the reporting of the results.

Thank you for your valuable evaluation and your time and we would like to respond to your comments.

- In the new Figure 3B the ROC curve of LUS score is not visible, moreover the authors state "Moreover, both LUS and severity predicting scores can significantly well differentiate between the DPLD and healthy control groups (AUC= 0.903, CI95%= 0.816 – 0.990, p < 0.001 for both LUS and severity scores, Fig 3B) as a score ≥ 2 had a sensitivity of 74.2% and 77.4% respectively with a specificity of 100% in predicting DPLD." This is not clear: do the severity score and LUS have the same AUC? then why the ROC of LUS is not reported in the figure?

Response. We agree with the reviewer that this point carries some confusion. Yes, the severity score and LUS have the same AUC; so, the ROC of LUS is superimposed on that of the severity score resulting in one curve. The similarity of AUC for both scores in the figure 3B is expected as the comparison is carried out between the healthy population and those with DPLD. Accordingly, in order to presented the ROC curve of both scores clearly, we separated the ROC of LUS and severity scores into 2 figures (shown below) to clarify this point. The modified figures are included in the manuscript as Fig 4A and 4B respectively (instead of Fig 3B “removed”) with correction of the text in the manuscript (line 246) and the legend of figure 3 (lines 254 - 257).

- I strongly encourage the authors to revise the manuscript with the help of a native English speaker, as many sections are still difficult to read and appear verbose or convoluted.

Response. Following the reviewer’s suggestion, a native English speaker reviewed the whole manuscript and grammatical mistakes / sentences were corrected (marked throughout the document).

Reviewer #2: The authors have adequately addressed the points raised and the overall quality of the manuscript is improved.

Thank you for your valuable evaluation and your time.

---

## [Decision Letter · Decision Letter 2]

26 Mar 2025

Oxygen desaturation and lung ultrasonography as markers of diffuse parenchymal lung diseases severity

PONE-D-24-24333R2

Dear Dr. Shafiek,

We’re pleased to inform you that your manuscript has been judged scientifically suitable for publication and will be formally accepted for publication once it meets all outstanding technical requirements.

Kind regards,

Vincenzo Lionetti, M.D., PhD

Academic Editor

PLOS ONE

Additional Editor Comments (optional):

Reviewers' comments:

Reviewer's Responses to Questions

**Comments to the Author**

1. If the authors have adequately addressed your comments raised in a previous round of review and you feel that this manuscript is now acceptable for publication, you may indicate that here to bypass the “Comments to the Author” section, enter your conflict of interest statement in the “Confidential to Editor” section, and submit your "Accept" recommendation.

Reviewer #1: (No Response)

2. Is the manuscript technically sound, and do the data support the conclusions?

Reviewer #1: Yes

3. Has the statistical analysis been performed appropriately and rigorously? 

Reviewer #1: I Don't Know

4. Have the authors made all data underlying the findings in their manuscript fully available?

Reviewer #1: Yes

5. Is the manuscript presented in an intelligible fashion and written in standard English?

Reviewer #1: Yes

6. Review Comments to the Author

Reviewer #1: No further comments. The authors have adequately replied to the points raised with the last round of review.

7. PLOS authors have the option to publish the peer review history of their article (what does this mean? ). If published, this will include your full peer review and any attached files.

**Do you want your identity to be public for this peer review?** For information about this choice, including consent withdrawal, please see our Privacy Policy .

Reviewer #1: No

---

## [Editor Report · Acceptance letter]

PONE-D-24-24333R2

PLOS ONE

Dear Dr. Shafiek,

I'm pleased to inform you that your manuscript has been deemed suitable for publication in PLOS ONE. Congratulations! Your manuscript is now being handed over to our production team.

Kind regards,

on behalf of

Prof. Vincenzo Lionetti

Academic Editor

PLOS ONE